# Measurement of Form and Position Error of Small-Diameter Deep Holes Based on Collaboration Between a Lateral Confocal Displacement Sensor and Helical Scanning

**DOI:** 10.3390/s25226863

**Published:** 2025-11-10

**Authors:** Yao Liu, Daguo Yu, Huifu Du, Tong Chen

**Affiliations:** School of Mechanical Engineering, North University of China, Taiyuan 030051, China; yudaguo@nuc.edu.cn (D.Y.); b20230211@st.nuc.edu.cn (H.D.); bg20240116@st.nuc.edu.cn (T.C.)

**Keywords:** spectral confocal, virtual slicing–B-spline reconstruction, small-diameter deep hole, helical scanning

## Abstract

In this study, an innovative measurement method integrating lateral confocal technology and composite motion control is proposed to address the physical space constraints and data processing problems in the detection of the shape and position errors in deep holes with large aspect ratios and small diameters. By designing a lateral confocal displacement sensor and a cantilever measuring device, we break through the spatial constraints of ∅6 mm
deep-hole inspection and solve the problems of rigidity and surface damage in the traditional contact probe. We constructed an axis-rotation coordinated motion control model and found that the measuring points were densely arranged in a helical trajectory along the inner wall of the hole. We developed the “virtual slicing–B-spline reconstruction” algorithm and used the adaptive motion control algorithm to achieve a more efficient measurement of the hole. The innovative “virtual slicing–B-spline reconstruction” algorithm, using adaptive grouping, dynamic slicing, and a fourth-order B-spline-fitting hierarchical processing framework, reached a straightness error assessment result of the 1 μm order. Experiments show that, under 0.5 mm∕s feed rate and 12 rmp rotational speed, the standard deviation of straightness is ≤0.0008 mm and the standard deviation of cylindricity is ≤0.0064 mm; compared to the CMM (coordinate measuring machine) measurement results, the cylindricity and straightness evaluation errors obtained by the new measurement method are reduced by 4.6% and 4.5%, respectively. It provides a technical solution that improves both accuracy and efficiency for the precision inspection of small-diameter deep holes.

## 1. Introduction

With the development of precision and high-end modern manufacturing, precision measurement technology has become one of the key indicators to measure a country’s manufacturing level. In the field of high-end equipment manufacturing, large-aspect-ratio (≥20) small-diameter (≤10 nm) deep hole structures are the core carrier of fluid transmission, precision positioning, and energy coupling, and their shape and position accuracy directly determines the performance of the product [1]. Taking the aerospace engine fuel nozzle as an example, for a ∅6 mm hole with an aspect ratio of 24, a 5 μm hole diameter error leads to an atomisation cone angle deviation of ±2° and increases the Sauter mean diameter (SMD) by 15%, which directly affects the engine thrust and service life [2,3]. Similarly, a hydraulic valve body deep bore case showed that 5 μm bore error resulted in a 15% increase in flow resistance but no 10% increase in oil line pressure loss, resulting in a system response delay of 8 ms, while a 1 μm straightness error equated to a local bore reduction of 0.5% [4,5].

Current precision measurement of the internal diameter of large-aspect-ratio small-diameter deep holes is faced with physical space limitations caused by difficulties around full-area coverage and the low efficiency of ultra-dense point cloud processing, which need to be resolved urgently. Existing measurement methods are mainly classified into contact measurement and non-contact measurement [6]. Contact measurement often uses direct contact between the probe and the measured hole wall to record the surface characteristics of the inner diameter of the deep hole [7]. There are two main limitations to the use of contact measurement methods for the measurement of large length-to-diameter ratio deep holes: its physical dimensions mean that the probe is affected by its rigid structural design and it is difficult to satisfy the high demand of large length-to-diameter ratio deep holes [8], and, at the same time, the probe’s cantilever beam structure produces significant flexural deformation under the action of gravity and contact reaction force, which leads to serious distortion of the measurement data [9]. Because of the limitations of contact measurement, the measurement process of the probe needs to exert a constant contact force in order to ensure the stability of the measurement, and it is easy to leave a typical indentation on the surface of the hole wall that has a high degree of finish, which fundamentally clashes with the high degree of finish of the inner surface of the deep hole [10].

Non-contact measurement, based on optical principles [11], allows us to directly obtain the 3D point cloud of the measured surface, without physical contact with the measured surface, breaking through the size and rigidity limitations of the contact probe, effectively circumventing the inherent defects of contact measurement, and it is suitable for detecting the form and position errors of large-diameter deep holes [12]. However, limited by the spatial constraints of small-diameter deep holes and the physical characteristics of the optical measurement system, in order to achieve the detection of small-diameter deep holes with full-area coverage, a spiral scanning path is used to generate an ultra-dense point cloud [13]. Existing algorithms for the processing of ultra-dense point clouds generally adopt strategies such as uniformly reduce sampling and fixed meshing to deal with such data [14], which makes it difficult to take into account the contradiction between the measurement accuracy and computational efficiency, a contradiction that is particularly prominent under the ISO 12780-1 standard [15], which requires form and position error assessment to be based on the geometrical characteristics of the complete contour. Therefore, there is an urgent need to develop a new algorithm that combines high-density point cloud feature extraction, computational efficiency, and evaluation accuracy to enable the performance of non-contact measurement in the inspection of deep holes with a large aspect ratio and small diameter.

Spectral confocal technology originates from the confocal microscopy proposed by the American scholar Minsky [16], and its core advantage lies in realizing absolute distance measurement through direct wavelength–distance mapping; which breaks through the speed bottleneck of traditional scanning measurement in principle. The sensor based on spectral confocal technology is a new type of high-precision, non-contact sensor that has emerged in recent years, and the theoretical measurement accuracy can reach the nanometer scale [17]; its core working principle is that composite light from a broad-spectrum light source passes through a beam-splitting prism and enters a dispersive objective lens, where it is split into monochromatic light of different wavelengths, forming gradient-focused color spots along the optical axis [18]. When the measured surface is at the focal point of a specific wavelength, the reflected light of this wavelength returns back in the original way and arrives at the spectrometer, which converts the wavelength information into the amount of physical displacement by identifying the spectral peak wavelength and combining it with the pre-calibrated wavelength–displacement mapping curve [19]. This non-contact measurement mode encodes spatial position information through wavelengths, circumventing in principle the speed problem of conventional scanning techniques. Due to the spectral confocal displacement sensor’s low requirements on the measured surface condition, allowing a larger tilt angle of the measured surface, fast measurement speed, and high real-time performance, it has rapidly become a popular sensor for industrial measurements and is widely used in the fields of precision positioning [20], thin-film thickness measurement [21], and precision measurement of microscopic contours [22].

Based on the advantages of spectral confocal technology, this paper proposes an innovative measurement method integrating lateral confocal spectral confocal technology and composite motion control. Through the design of lateral confocal sensors and cantilever measurement devices, the measurement points are arranged along the inner wall of the deep hole in a helical and dense trajectory, which solves the problem of the inability to achieve full-area coverage due to physical space limitations in the detection of small-diameter deep holes. At the same time, based on the non-uniform rational B-spline theory, the innovative design of the ‘virtual slicing–B-spline reconstruction algorithm’ is used to achieve dimensionality reduction and reconstruction in the point cloud data through the progressive processing of the topological grouping of the measurement point set, spatial fitting, multi-layer virtual slicing, and least-squares error evaluation, so as to solve the problem of high-efficiency processing of the super-dense point cloud for the detection of small-diameter deep holes. Experimentally, it is proven that the measurement device and calculation method can effectively measure and evaluate the internal diameter of small-diameter deep holes with a large aspect ratio.

## 2. Measurement Principle

For the structural characteristics of the large-aspect-ratio small-diameter deep hole, based on the lateral confocal technology of a small-diameter deep-hole part detection device for targeted mechanical structure design, as shown in Figure 1. The structure fully meets the actual needs of small-diameter deep-hole part detection: the measured small-diameter deep-hole test piece is clamped by the clamping device, the probe rod is not in contact with the inner wall of the deep hole, and it is mounted in a fixed position on the magnetic levitation linear motor. A spectral confocal displacement sensor probe is mounted on the free end of the probe rod. The servo motor drives the clamping device and the small-diameter deep-hole test piece under test to rotate, and the high-precision magnetic levitation linear motor drives the spectral confocal displacement sensor probe to move along the axis of the test piece, which establishes an axial–rotational synergistic motion mechanism, forming a complete set of inspection motion systems.

The measurement radius R for a small-diameter deep hole with a large aspect ratio in Figure 1 is derived from the measurement radius r of the spectral confocal displacement sensor and the deviation ∆s between the probe axis and the axis of the deep hole being measured. The specific mathematical model is as follows:(1)R=r+∆s

The measurement principle of the lateral confocal spectral confocal displacement sensor is shown in Figure 2. The white light source S sends out broad-spectrum visible light, which passes through pinhole $P_{1}$ to form an approximate point light source; it passes through the beam-splitting prism into the dispersion objective lens, where it is divided into different wavelengths of monochromatic light. An optical mirror adjusts the light path, causing the focal spots to distribute vertically along the axis. According to the light path direction and geometric relationships, only the monochromatic light of a specific wavelength t ($\lambda_{2}$), whose focal point aligns with the specimen surface, can return along the original path, pass through the dispersive objective lens and beam-splitting prism, and reach the spectrometer via pinhole $P_{2}$; other wavelengths of monochromatic light cannot be focused at this point, as the formation of diffuse spots means that they cannot enter the spectrometer. The spectrometer analyses the narrow-band spectrum and, based on the pre-calibrated wavelength–displacement mapping curve, obtains the absolute distance of the measured surface from the axis of the dispersion objective.

Using the axis-rotation synergistic motion mechanism to drive the parts to be tested and the lateral spectral confocal displacement sensor movement to complete the detection of large-aspect-ratio small-diameter deep holes, we elucidated the presentation of the helical path of the ultra-dense point cloud $P_{i}$ along the small-diameter deep hole in the wall and the spatial coordinates of the cloud of points $P_{i}\left(x_{i}{,y}_{i},z_{i}\right)$, which are shown in Figure 3.

## 3. Compensation Methods

### 3.1. Data Preprocessing

Spectral confocal displacement sensors are used in high-precision non-contact measurement equipment; their measurement accuracy can reach the sub-micron level. In the process of measuring the bore diameter of a small-diameter deep hole, due to complex measurement environments (uneven local illumination, temperature drift, mechanical micro-vibration, etc.) and the reflective interference of the hole wall, the collected data will inevitably contain outliers and non-structural noise, which will seriously interfere with the quality of the collected points and interfere with the subsequent geometric fitting and structural analysis, so the first step is to select an appropriate filtering algorithm to process noise after obtaining the collected data points.

Non-structural noise caused by temperature drift, mechanical micro-vibrations, and other factors during data acquisition is filtered out by setting acquisition peak height thresholds and sharpness thresholds. Remaining noise components are eliminated using wavelet threshold filtering [23]. Firstly, based on the scale function $\phi_{J,k}(t)$ and wavelet function $\psi_{j,k}(t)$ of the db4 wavelet function, combined with the low-frequency approximation coefficients $c_{J,k}$ and high-frequency detail coefficients $d_{j,k}$, the original signal r(t) is decomposed via J-layer decomposition, and the signal is unfolded into the multiresolution expression form:(2)$$r(t)=\sum c_{J,k}\phi_{J,k}(t)+\sum_{j=1}^{J}\sum_{k}d_{j,k}\psi_{j,k}(t)$$

Based on the differences in the characteristics of measurement signals and noise in the wavelet domain, SURE (Stein’s Unbiased Risk Estimate) adaptive soft thresholding is used for the high-frequency $d_{j,k}$ coefficient of each layer to perform detailed threshold quantization processing. The noise standard deviation $\sigma_{j}$ of this layer is the median absolute deviation, so the threshold $T_{j}$ satisfies the following:(3)$$T_{j}=arg\;\underset{T}{\min}\left[N_{j}\sigma_{j}^{2}+\sum_{k=1}^{N_{j}}min{\left(\left|d_{j,k}\right|,T\right)}^{2}-2\sigma_{j}^{2}\cdot I\left(\left|d_{j.k}\right|<T\right)\right]$$

For the signal after the noise reduction process, with the help of the inverse wavelet transform, the coefficients of each part of the process are reassembled to create a shape close to the original real signal, to achieve signal reconstruction after removing noise interference. The inverse wavelet transform reconstructs the following:(4)$$\hat{r}\left(t\right)=\sum_{k}c_{J,k}\phi_{J,k}(t)+\sum_{j=1}^{J}\sum_{k}{\hat{d}}_{j,k}\psi_{j,k}(t)$$

### 3.2. Virtual Slicing–B-Spline Reconstruction Method

The virtual slicing–B-spline reconstruction method is innovatively designed to rectify the point cloud extraction distortion, low efficiency of form, and errors in position evaluation faced by existing algorithms when dealing with ultra-dense point clouds in helical paths, based on the helical density characteristics of point clouds in spectral confocal measurement and combined with the geometrical accuracy inspection requirements of deep holes with a large L/D ratio. Through the hierarchical processing architecture of ‘point cloud grouping-curve fitting-slice reconstruction-error solving’, the method converts the ultra-dense point cloud generated by the helical scanning path into regular cross-section data and then obtains the shape and position of deep holes with a large L/D ratio and small diameter accurately and efficiently. The shape and position error of large-length-to-small-diameter deep holes can be accurately and efficiently obtained. The specific calculation flow of the virtual slice–B-spline reconstruction algorithm is as follows:

(1) After the wavelet threshold filtering preprocessing of the measured point cloud $\left\{P_{i}\right\}\left(1,2,\cdots ,N\right)$, the least squares cylindrical fitting method [24] is used to fit the axis, and the theoretical axis is a straight line in space $L_{0}$, and the direction vector is $V_{0}$.

(2) Based on the circumferential position information of each point in the point cloud data, according to the angle of any two points, between the angle difference of 2π, integer multiples of the point are divided into M groups of point sets $S_{j}=\left\{P_{jk}\right\}\left(j=1,2,\cdots ,M\right)$. The number of groups M is jointly determined by the rotational speed ω of the measured workpiece and the sampling period T of the spectral confocal displacement sensor, M=2π/ωT, while subgroups need to be considered subgroups of the angular interval. For the first group of j, corresponding to the angular interval θ, the following formula should be satisfied:(5)$$\theta =\left[\frac{2\pi \left(j-1\right)}{M}-\frac{\pi }{M},\frac{2\pi \left(j-1\right)}{M}+\frac{\pi }{M}\right]$$

(3) Each point set $S_{j}$ is fitted along the axis direction using a B-spline curve (Basis Spline) [25]. By calculating the basis function $N_{i,k}\left(u\right)$ and control point $P_{i}$, the expression for generating continuous spatial curves is(6)$$C_{j}\left(u\right)=\sum_{i=0}^{n}P_{i}N_{i,k}\left(u\right),u\in \left[u_{k}{,u}_{n+1}\right]$$

(4) A series of slice planes $\Pi_{k}$ perpendicular to the vector $V_{0}$ are defined along the direction of the initial fitting axis $L_{0}$ from the point cloud axial minimum projection $z_{min}^{\prime }$ to the maximum projection $z_{max}^{\prime }$ at equal spacing ∆L with the equations:(7)$$\Pi_{k}∶L_{\Pi }=L_{\Pi_{k}}=z_{min}^{\prime }+k∆L,\left(k=0,1,\cdots ,K\right)$$

(5) For each slice plane $\Pi_{k}$, we calculate the intersection with all curves $C_{j}\left(u\right)$ as ${Q_{jk}=C}_{j}\left(z_{k}\right)$ by numerical iteration to form a new intersection point set $\left\{Q_{jk}\right\}$.

(6) For the intersection point set $\left\{Q_{jk}\right\}$ on each plane $\Pi_{k}$, the least squares method [26] is used to fit the circle $\Gamma_{k}$, in order to obtain the centre of each fitted circle. The specific operation is as follows:

(a) Neglecting the axial coordinates in the $\Pi_{k}$ plane and defining the local coordinate system $\left(x^{\prime },y^{\prime }\right)$, let the coordinates of the centre of the circle be $\left(x_{c}^{\prime },y_{c}^{\prime }\right)$ and the radius be $R^{\prime }$, then the equation of the circle in the plane is expressed as(8)$${\left(x^{\prime }+x_{c}^{\prime }\right)}^{2}+{\left(y^{\prime }+y_{c}^{\prime }\right)}^{2}=R^{\prime 2}$$

(b) The objective function is the minimum of the sum of the squares of the algebraic distances from a point to a circle, and the mathematical model is(9)$$\underset{x_{c}^{\prime },y_{c}^{\prime },R^{\prime }}{\min}\sum_{j=1}^{M}{\left({\left(x^{\prime }+x_{c}^{\prime }\right)}^{2}+{\left(y^{\prime }+y_{c}^{\prime }\right)}^{2}-R^{\prime 2}\right)}^{2}$$

(c) Construct a system of linear equations and introduce auxiliary variables:(10)$$H=x_{c}^{\prime 2}+y_{c}^{\prime 2}-R^{\prime 2}$$

(d) Then for each intersection point ${\hat{Q}}_{\mathrm{jk}}=\left(x_{j}^{\prime },y_{j}^{\prime }\right)$, construct the following equation:(11)$$\left[\begin{array}{ccc} \begin{array}{c} 2x_{1}^{\prime }\\ 2x_{2}^{\prime }\\ \begin{array}{c} \vdots \\ 2x_{M}^{\prime } \end{array} \end{array} & \begin{array}{c} 2y_{1}^{\prime }\\ 2y_{2}^{\prime }\\ \begin{array}{c} \vdots \\ 2y_{M}^{\prime } \end{array} \end{array} & \begin{array}{c} -1\\ -1\\ \begin{array}{c} \vdots \\ -1 \end{array} \end{array} \end{array}\right]\left[\begin{array}{c} x_{c}^{\prime }\\ y_{c}^{\prime }\\ H \end{array}\right]=\left[\begin{array}{c} x_{1}^{\prime 2}+y_{1}^{\prime 2}\\ x_{2}^{\prime 2}+y_{2}^{\prime 2}\\ \begin{array}{c} \vdots \\ x_{M}^{\prime 2}+y_{M}^{\prime 2} \end{array} \end{array}\right]$$

(e) Solving for this by means of the regular equation yields(12)$$\left\{\begin{array}{c} x_{c}^{\prime }=\frac{\sum x_{j}^{\prime }\left(x_{j}^{\prime 2}+y_{j}^{\prime 2}\right)-\frac{1}{M}\sum x_{j}^{\prime }\sum \left(x_{j}^{\prime 2}+y_{j}^{\prime 2}\right)}{\sum x_{j}^{\prime 2}+\sum y_{j}^{\prime 2}-\frac{1}{M}{\left(\sum x_{j}^{\prime }\right)}^{2}-\frac{1}{M}{\left(\sum y_{j}^{\prime }\right)}^{2}}\\ y_{c}^{\prime }=\frac{\sum y_{j}^{\prime }\left(x_{j}^{\prime 2}+y_{j}^{\prime 2}\right)-\frac{1}{M}\sum y_{j}^{\prime }\sum \left(x_{j}^{\prime 2}+y_{j}^{\prime 2}\right)}{\sum x_{j}^{\prime 2}+\sum y_{j}^{\prime 2}-\frac{1}{M}{\left(\sum x_{j}^{\prime }\right)}^{2}-\frac{1}{M}{\left(\sum y_{j}^{\prime }\right)}^{2}}\\ H=\frac{1}{M}\sum (x_{j}^{\prime 2}+y_{j}^{\prime 2})-2x_{c}^{\prime }\cdot \frac{1}{M}\sum x_{j}^{\prime }-2y_{c}^{\prime }\cdot \frac{1}{M}\sum y_{j}^{\prime } \end{array}\right.$$

(f) Obtain the centre point set $\left\{Q_{k}\right\}$ of the fitted circle for all cross-sectional intersection points.

(7) Taking the set of fitted circle-centred points $\left\{Q_{k}\right\}$ obtained from all calculations as data samples, the straightness error of the measured point cloud is calculated based on the minimum inclusive cylindrical optimization method [24] for circle-centred trajectories. Construct the spatially constrained optimisation model of the axis: set the axis as a spatial straight line $L_{axis}$ over the point $A=\left(a_{x}{,a}_{y}{,a}_{z}\right)$, direction vector $V=\left(V_{x}{,V}_{y}{,V}_{z}\right)$ (unit vector $\left\|V\right\|=1$), and radius r and establish the following nonlinear planning equation:(13)$$\left\{\begin{array}{c} \underset{A,V,r}{\min}r\\ s.t.\\ \begin{array}{c} \left\|\left(Q_{k}-A\right)\times V\right\|\leq r,\forall k=1,2,\cdots ,K\\ V_{x}^{2}+V_{y}^{2}+V_{z}^{2}=1 \end{array} \end{array}\right.$$

The expression for the gradient of the constraint $\left\|\left(Q_{k}-A\right)\times V\right\|\leq r$ is(14)$$\nabla \left(\left\|\left(Q_{k}-A\right)\times V\right\|-r\right)=\left[\begin{array}{c} -\frac{\left(Q_{k}-A\right)\times V\times V}{\left\|\left(Q_{k}-A\right)\times V\right\|}\\ \frac{\left(Q_{k}-A\right)\times \left(Q_{k}-A\right)\times V}{\left\|\left(Q_{k}-A\right)\times V\right\|}\\ -1 \end{array}\right]$$

The model takes the minimum cylindrical radius that accommodates all the coordinates of the centre of the circle as the optimisation objective and is solved iteratively using Sequential Quadratic Programming (SQP) [27] and nonlinear constrained gradient optimisation to finally obtain the minimum cylindrical radius $r_{min}$, then the straightness error $\delta_{L}$ of the tubing to be measured is(15)$$\delta_{L}=2r_{min}$$

(8) For the measurement point cloud $\left\{P_{i}\right\}\left(1,2,\cdots ,N\right)$, after wavelet filtering preprocessing, based on the minimum external inclusive cylinder optimisation method of the spiral dense row trajectory of the measurement point cloud, with the minimum external cylindrical column radius inclusive of all the measurement point clouds as the optimisation objective, sequential quadratic programming (SQP) and nonlinear constrained gradient optimisation are used for iterative solving to ultimately obtain the minimum external cylindrical column radius $R_{outer}$ and establish the nonlinear planning equation as(16)$$\left\{\begin{array}{c} \underset{{A,V,R}_{\mathrm{outer}}}{\min}R_{outer}\\ s.t.\\ \begin{array}{c} \left\|\left(Q_{jk}-A\right)\times V\right\|\leq R_{outer},\forall Q_{jk}\\ \left\|V\right\|=1 \end{array} \end{array}\right.$$

Based on the obtained minimum circumcircle cylinder axis, we use vector cross-multiplication to calculate the distance from all measurement points to the axis. We draw a cylinder passing through the measurement point with the minimum distance, and the difference in radius between the two cylinders is the cylindricity of the minimum circumcircle cylinder optimization method, denoted as $\delta_{R_{outer}}$.

Similarly, for the measurement point cloud $\left\{P_{i}\right\}\left(1,2,\cdots ,N\right)$, the maximum inscribed cylinder optimization method is adopted, taking the maximum inscribed cylinder radius that excludes all measurement points as the optimization objective. Iterative solutions are performed using Sequential Quadratic Programming (SQP) and nonlinear constrained gradient optimization to finally obtain the maximum inscribed cylinder radius $R_{inner}$, and the nonlinear programming equation is established as(17)$$\left\{\begin{array}{c} \underset{{A,V,R}_{\mathrm{inner}}}{\min}R_{inner}\\ s.t.\\ \begin{array}{c} \left\|\left(Q_{jk}-A\right)\times V\right\|\geq R_{inner},\forall Q_{jk}\\ \left\|V\right\|=1 \end{array} \end{array}\right.$$

Taking the axis of the obtained maximum inscribed cylinder as the reference, the vector cross-multiplication method is used to calculate the distances from all measurement points to the axis. A cylinder is drawn through the measurement point with the maximum distance, and the radius difference between the two cylinders is the cylindricity derived from the maximum inscribed cylinder optimization method, denoted as $\delta_{R_{inner}}$. By comparing $\delta_{R_{outer}}$ and $\delta_{R_{inner}}$, the cylindricity error $\delta_{R}$ of the pipe to be measured is determined as follows:(18)$$\delta_{R}=min\left(\delta_{R_{outer}},\delta_{R_{inner}}\right)$$

## 4. Experimental Verification

### 4.1. Lab Bench Construction

According to the above measurement principle, an experimental bench for small-diameter deep-hole detection was built, as shown in Figure 4. The magnetic levitation plate linear motor drives the probe rod, driving the spectral confocal displacement sensor probe along the axial movement of the deep hole; the servo motor drives the circumferential rotation of the measured test piece, to establish the spatial trajectory optimisation mechanism of the axial–rotary cooperative movement and to realise the measurement of points in the inner wall of the deep hole in the helical dense rows of the layout. T700 carbon (Jiangxi Shanmeibang Fiber Products Co., Ltd., Ji’an, China) fibre tubes are used for the support rods, with an inner diameter of 4 mm, an outer diameter of 5 mm, and a free cantilever length of 350 mm. The selected spectral confocal displacement sensor is the CRL1500N model (Lingguang Automation Technology Co., Ltd., Yueqing, Zhejiang Province, China), featuring a reference distance of 3 mm, a measurement range of ±0.75 mm, a spot diameter of ∅17 μm, an outer diameter of ∅3.8 mm, and a weight of 23 g.

When the probe assembly enters the interior of the tube to be measured along the axis, as shown in Figure 1, in order to accurately capture the position of the hole wall and ensure the feasibility and accuracy of the measurement, the minimum value of the hole diameter to be measured $D_{min}$, the outer diameter of the support rod D, and the minimum value of the effective range of the spectral confocal displacement sensor $r_{min}$ need to satisfy certain design criteria. Meanwhile, to avoid mechanical interference during the measurement and increase the safety clearance δ during movement, the diameters of all large-aspect-ratio deep holes measured using this device should satisfy the relationship shown in Equation (19):(19)$$D_{min}\geq r_{min}+\frac{D}{2}+\delta$$

The actual measurement of the measuring device, because the probe is driven by a linear motor to feed at speed v, the measured test piece rotates with the servomotor at angular speed ω. The coupling of the two motions places the probe in a non-inertial reference system, which generates dynamic loads such as Coriolis force and centrifugal inertia force, so the motion compensation is taken into account while considering the deflection of the cantilever, and the mathematical model of the overall compensation is established.(20)$$y_{total}=\frac{64\left[mg+\omega^{2}\Delta s\left(m+\frac{1}{2}m_{tube}\right)+2\omega v\sin\theta \left(m+\frac{1}{2}m_{tube}\right)\right]L^{3}}{3\pi E\left(D^{4}-d^{4}\right)}$$

m: mass of the probe (g); $m_{tube}$: mass of the carbon fibre tube(g); ∆s: radial distance from the probe to the workpiece’s rotational axis (mm); L: effective length of the cantilever beam (mm); E: elastic modulus of the carbon fibre tube (≥200 GPA); D: outer diameter of the carbon fibre tube (mm); d: inner diameter of the carbon fibre tube (mm); θ: angle between the motion direction and the rotation plane (°).

### 4.2. Measurement Results

Set the feed speed of the magnetic levitation motor as 0.5 mm∕s, the rotation of the test piece to be measured driven by the servo motor as 12 rpm, and the sampling period of the spectral confocal displacement sensor as T=0.1 s. Combined with the parameters of the measuring device, according to the formula (20), we calculate the total compensation $y_{total}=0.889\;mm$ for the probe assembly part of the measuring device in the measuring process, and take the safety gap δ=1 mm, and according to the formula (19), we get the minimum deep-hole size that the measuring device can measure, which is ∅5.75 mm, with the maximum length of 250 mm.

Now, practical measurements are conducted on a ∅6×200 pipe fitting specially fabricated for this study using a measurement device. After the acquired measurement data is denoised by wavelet threshold filtering, the geometric and positional errors of the pipe fitting to be measured are obtained by adopting the virtual slicing–B-spline reconstruction method. The specific process is as follows:

For the point cloud acquired by the lateral confocal spectral confocal displacement sensor, wavelet threshold filtering is used to implement noise reduction processing, Figure 5 presents the distribution comparison of the point cloud before and after noise reduction using wavelet threshold filtering: the blue curve represents the signal before noise reduction, and the red curve represents the signal after noise reduction by radial wavelet threshold filtering. As can be seen from the figure, the fluctuation in the point cloud data is more stable after processing, effectively weakening the noise interference, showing the good effect of wavelet threshold filtering in point cloud noise reduction.

According 12 rpm of the measured test piece driven by the servo motor and the sampling period T=0.1 s of the spectral confocal displacement sensor, the filtered and denoised point cloud is grouped into M=50 groups with an angular interval of θ=7.2° per group. The spatial curve fitting of each point set is completed by a fourth-order B-spline curve, and the fitting results are shown in Figure 6.

Orthogonal virtual slices are established at an equal spacing of ∆L=1 mm to obtain the intersection points of the virtual slices with all spatial spline curves. In the plane of the virtual slices, a circle is fitted to these intersection points using the least squares method, yielding the coordinates of the fitted circle’s centre within the slices, as shown in Figure 7.

According to the ISO 12780-1 standard for the evaluation of spatial straightness error, the minimum inclusive cylindrical method for fitting the set of centroids obtained by calculating the straightness error of the measurement of the large length-to-diameter ratio of a small-diameter deep hole is $\delta_{L}=0.0106\;mm$, as shown in Figure 8.

According to the measured point cloud $\left\{P_{i}\right\}\left(1,2,\cdots ,N\right)$ along the spiral dense-row trajectory on the inner wall of the small-diameter deep hole, with the aid of a data analysis program developed based on MATLAB R2024b, the cylindricity results of the minimum circumscribed cylinder optimization method (MCC) and the maximum inscribed cylinder optimization method (MIC) are compared. The cylindricity error of the deep hole in this measurement is obtained as $\delta_{R}=0.1206\;mm$ from the maximum inscribed cylinder optimization method, as shown in Figure 9.

#### 4.2.1. Repeated Measurement Results

Five measurements were conducted on the same test pipe section under varying feed rates, rotational angular velocities, and sampling periods. The straightness results for the inner diameter of the tube, measured using the virtual slice–B-spline reconstruction method, are shown in Table 1 below, and the cylindricity results are shown in Table 2 below.

Analysing the straightness measurement results, the standard deviation of straightness for all groups is ≤0.0008 mm, and the extreme deviation is ≤0.0019 mm, which indicates that the virtual slicing–B-spline reconstruction method maintains high repeatability under different motion parameters. At a fixed feed rate of 0.5 mm∕s, the rotational speed of the test tube increased from 4 rpm to 24 rpm with the straightness error variation ≤0.0006 mm. Given the structural characteristics of the test bench, the probe and the test piece remain separated during measurement. The probe axis aligns with the workpiece’s rotational axis during measurement, rendering the centrifugal inertial force exerted on the probe in a non-inertial reference frame negligible. Consequently, the increase in straightness error is insignificant. When the rotational speed is fixed at 24 rpm, the feed rate doubles and the straightness error increases by 7.8%. Combined with the structural characteristics of the experimental bench, the high-speed feed exacerbates the vibration of the cantilever beam structure, and the increase in the Coriolis force directly leads to an increase in the dynamic offset of the probe, which results in a decrease in the axial positioning accuracy of the probe and an increase in the straightness error. Variations in the sampling cycle directly impact the quantity of point clouds captured. However, from an overall analysis of straightness error, the assessment of straightness requires relatively low point cloud density. In scenarios where efficiency takes precedence, motion parameters may be appropriately increased.

Analysing the results of cylindricity measurements, the standard deviation of cylindricity for all groups is ≤0.0064 mm, and the extreme deviation is ≤0.0178 mm, which indicates that the virtual slicing–B-spline reconstruction method maintains a high reproducibility under different motion parameters. The increase in feed rate widens the axial sampling point spacing, causing the virtual slice distance ∆L to exceed the scale of local curvature variation within the deep hole. Concurrently, the reduction in sampling points leads to the omission of certain critical topographical features, directly impacting the calculation of the deep-hole cylindricity error. The increase in the rotation speed makes the sampling point angle interval increase, which leads to a decrease in the B-spline reconstruction accuracy and an increase in the cylindricity error. The sampling cycle directly influences the number of sampling points per unit time and the point cloud density. From the overall analysis of the cylindricity error, the evaluation of cylindricity is more dependent on the density of the full circumferential point cloud, so for some high-precision scenarios, a lower motion parameter should be used to complete the detection of the test piece.

Combining the results of straightness and cylindricity calculations for all groups, the virtual slicing–B-spline reconstruction method has high repeatability in straightness and cylindricity calculations.

#### 4.2.2. Measurement Result Verification

In order to verify the accuracy of the virtual slicing–B-spline reconstruction method, a fully automated coordinate measuring machine (CMM) NCA686 (Qingdao Leidun CNC Measuring Equipment Co., Ltd.Qingdao, Shandong Province, China) was used to make comparative measurements on the same ∅6×200 fittings.

Due to the structural constraints of the large L/D ratio of the pipe to be measured, the CMM cannot complete full-hole measurement in one operation and requires segmental measurement followed by overall fitting. Constrained by the structural characteristics of the large length-to-diameter ratio of the pipe section being tested, the coordinate measuring machine (CMM) cannot complete a full-bore measurement in a single operation. A 2mm ruby stylus and a 90° probe were selected. The CMM’s integrated RationDMIS (Dimensional Measuring Interface Standard) software was used to perform overall fitting of the two end cylinders. The cylindricity of the tubing measured by the CMM is 0.1269 mm, and the straightness of the inner diameter of the tubing is 0.0111 mm. A comparison of the results of the CMM and the straightness and cylindricity of the inner diameter of the tubing calculated by the virtual slicing–B-spline reconstruction method is shown in Table 3 below.

Through the comparative analysis of the measurement results, the straightness and cylindricity of the pipe fittings calculated by the virtual slicing–B-spline reconstruction method are close to the CMM results, and both of them meet the requirements of ISO 12780-1 for precision measurement. The deviation between the helical scanning and CMM results using the spectral confocal displacement sensor is mainly due to the differences in the measurement principle and data processing logic. Helical scanning is able to measure the inner diameter of deep holes continuously and with full coverage and directly obtain the three-dimensional point cloud data of the inner diameter of deep holes, which not only avoids the intermittent errors caused by artificial segmentation and improves the measurement speed but also, by virtue of the non-contact measurement characteristics, avoids the tiny indentation in the hole wall due to the contact force of the CMM probe and ensures the integrity of the measurement object; the virtual slicing–B-spline reconstruction method uses the fourth-order B-spline to fit the helical point cloud to the B-spline curve. The virtual slice–B-spline reconstruction method uses a fourth-order B-spline curve to fit the spiral point cloud, which can restore the true contour of the pipe more accurately than the default least squares method of the CMM, while the ‘overall fit’ of the CMM’s segmented measurement is essentially a linear interpolation, making it difficult to accurately reflect the local shape and positional tolerance of the intermediate region of the deep hole, while the full-hole coverage of the helical scanning can capture the details of the deep-hole features, achieving a highly accurate shape and positional tolerance. However, the full-bore coverage of helical scanning can capture the detailed features of deep holes, enabling highly accurate inspections of form and position tolerances.

## 5. Conclusions

Addressing the physical space limitations and data processing problems in detecting the internal diameter form error in small-diameter deep holes with large aspect ratios, a helical scanning measurement method based on lateral confocal technology is proposed, and the sub-micron precision detection of the form tolerance of small-diameter deep holes is realised by integrating innovation in detection device design, motion control optimization, and algorithm reconstruction. At the inspection device level, a cantilevered lateral confocal measurement mechanism is developed, which is driven by a carbon fibre support rod and a magnetically levitated linear motor to complete full-area coverage inspection of ∅6 mm deep holes and solve the problem of measurement failure caused by poor rigidity and surface damage with traditional contact probes; at the motion control level, a spatial trajectory optimization mechanism of axis–rotary cooperative motion is established by matching parameters between the magnetically levitated linear motor and the servo rotary motor, and measuring points are densely arranged in a helical pattern on the deep-hole inner wall. Experiments show that under a feed speed 0.5 mm∕s and a rotational speed of 12 rpm, the standard deviation of repeated straightness measurements is ≤0.0008 mm, and the standard deviation of repeated cylindricity measurement is ≤0.0064 mm, which meet the requirements of the ISO 12780-1 standard for full-contour data collection. At the algorithm level, the innovative ‘Virtual Slicing–B-Spline Reconstruction Algorithm’ is proposed, which realizes the efficient processing of an ultra-dense point cloud through the framework of adaptive grouping, dynamic slicing, and a fourth-order B-spline fitting hierarchy; the straightness evaluation error assessment reaches the 1.0 μm order, compared with CMM measurements and the cylindricity and straightness evaluation errors of the proposed method are reduced by 4.6% and 4.5% respectively. This improvement in accuracy is due to the double advantages of the measurement principle and data processing.

This study integrates spectral confocal displacement sensor technology and intelligent algorithms to break through the limitations of large-aspect-ratio and small-diameter deep-hole measurement, which is of great significance in promoting ultra-precision inspections and upgrading inspection technology in engineering.

## Figures and Tables

**Figure 1 sensors-25-06863-f001:**
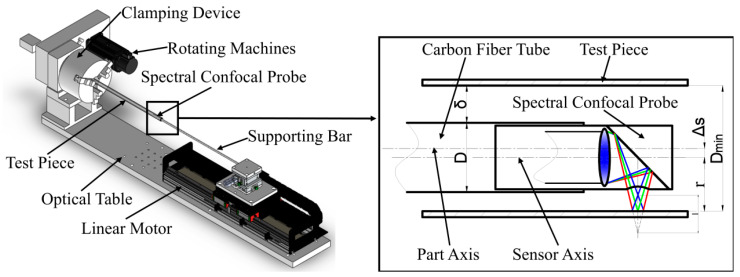
Three-dimensional model of a small-diameter deep-hole inspection apparatus based on lateral confocal principles.

**Figure 2 sensors-25-06863-f002:**
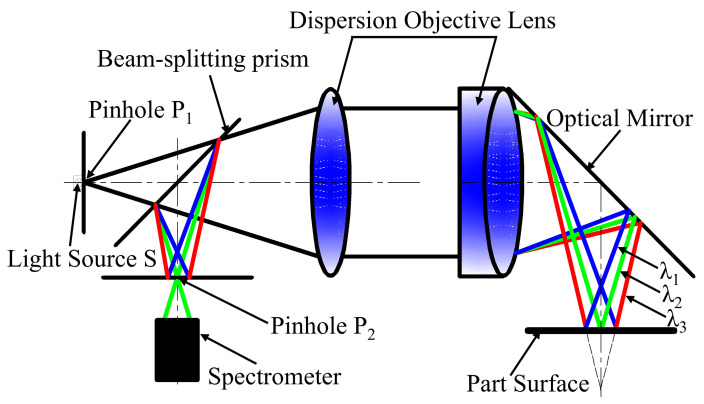
Lateral confocal spectrum: confocal displacement sensor measurement principle.

**Figure 3 sensors-25-06863-f003:**
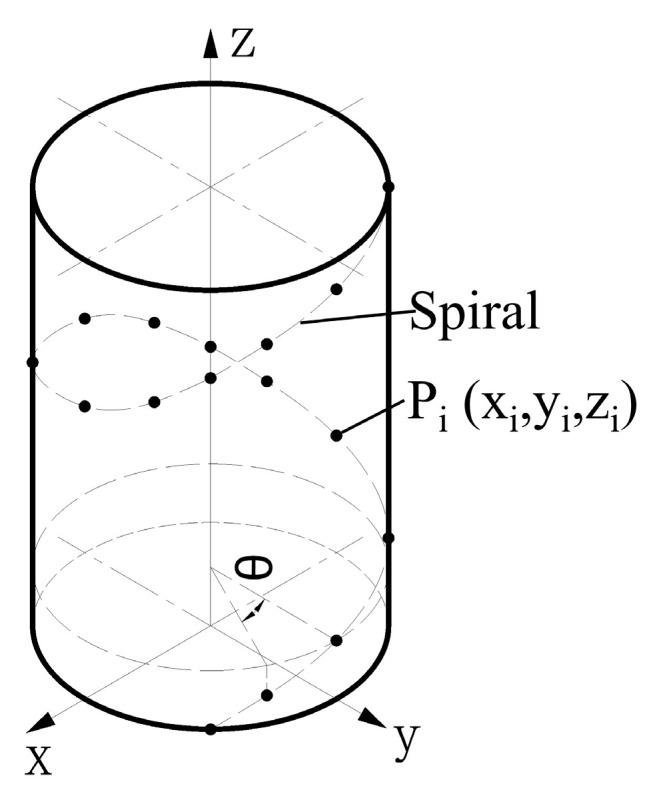
Schematic diagram of point cloud from helical scanning trajectory along the deep borehole inner wall.

**Figure 4 sensors-25-06863-f004:**
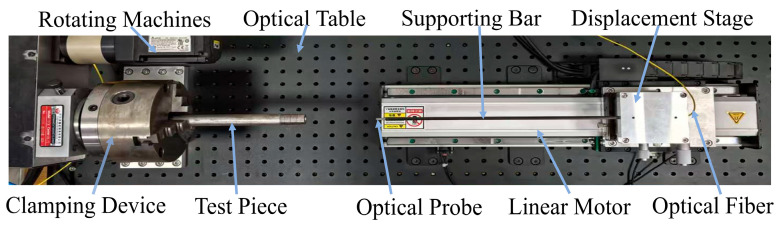
Physical diagram of deep-hole detection based on spectral confocal principle.

**Figure 5 sensors-25-06863-f005:**
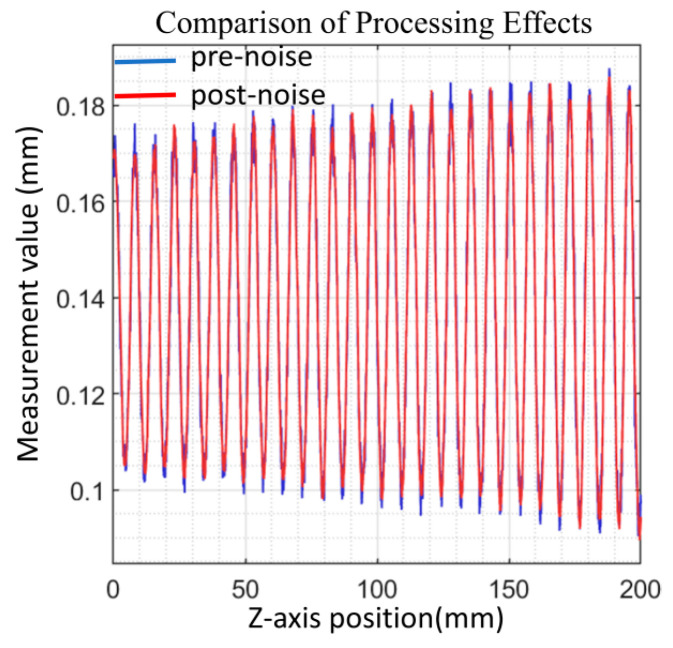
Wavelet thresholding for noise reduction processing.

**Figure 6 sensors-25-06863-f006:**
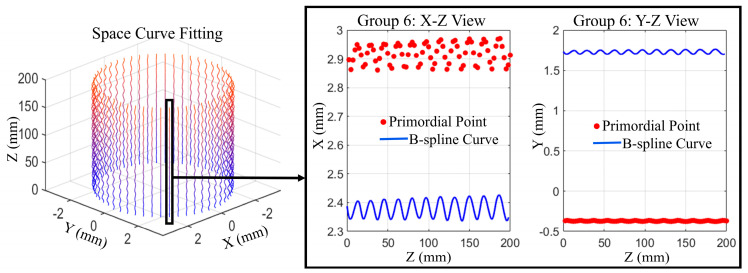
Virtual slice–B-spline reconstruction algorithm yielding fourth-order B-spline spatial curve fitting results.

**Figure 7 sensors-25-06863-f007:**
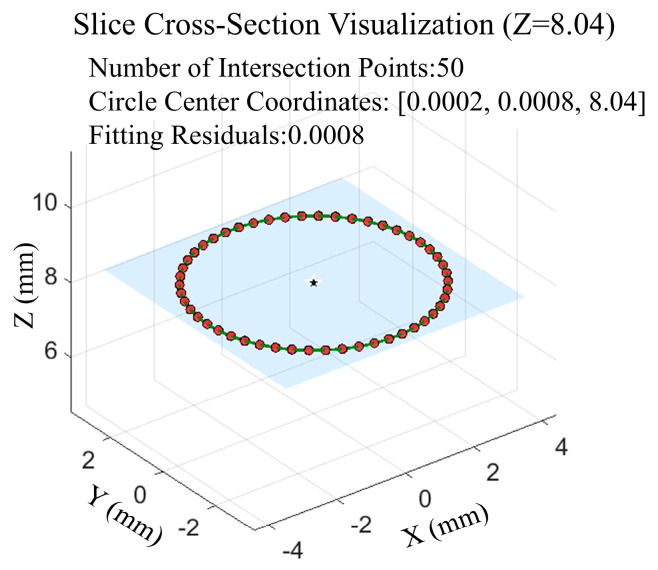
Intersection point with the B-spline curve within the virtual slice section (Z = 8.04 mm) and the least squares-fitted circle.

**Figure 8 sensors-25-06863-f008:**
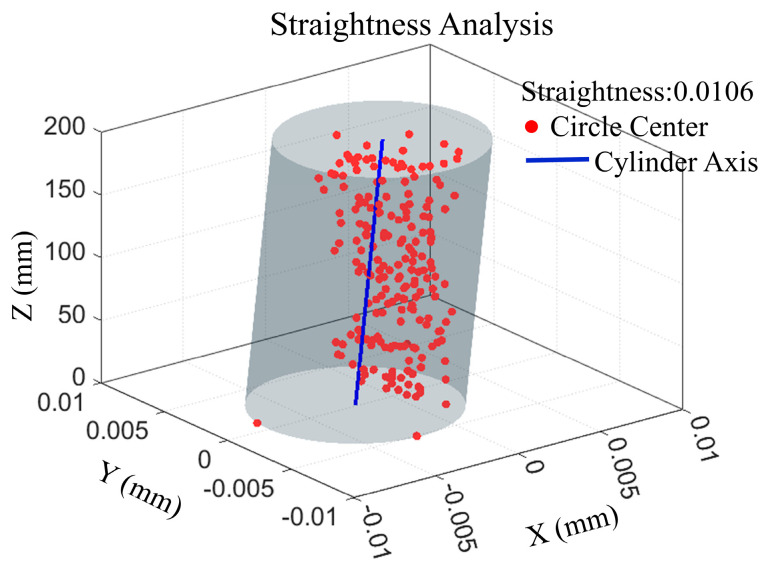
Straightness error analysis result.

**Figure 9 sensors-25-06863-f009:**
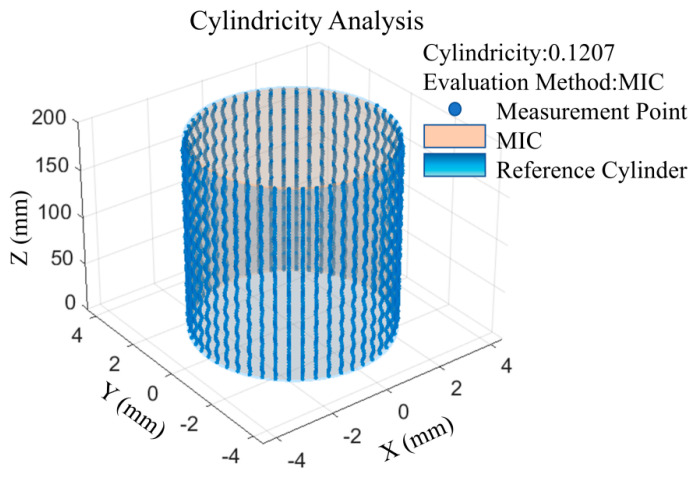
Cylindricity error analysis results.

**Table 1 sensors-25-06863-t001:** Measured straightness results (unit: mm).

Feed Rate (mm/s)	Rotation Speed (rpm)	Sampling Period (s)	First	Second	Third	Fourth	Fifth	Average Value
0.5	4	0.05	0.0100	0.0096	0.0094	0.0104	0.0088	0.0096
0.5	8	0.05	0.0095	0.0099	0.0087	0.0106	0.0091	0.0096
0.5	12	0.1	0.0106	0.0107	0.0092	0.0091	0.0096	0.0098
0.5	24	0.1	0.0100	0.0098	0.0111	0.0108	0.0092	0.0102
1	24	0.1	0.0115	0.0117	0.0100	0.0110	0.0106	0.0110

**Table 2 sensors-25-06863-t002:** Measured cylindricity results (unit: mm).

Feed Rate (mm/s)	Rotation Speed (rpm)	Sampling Period (s)	First	Second	Third	Fourth	Fifth	Average Value
0.5	4	0.05	0.1167	0.1211	0.1123	0.1189	0.1145	0.1167
0.5	8	0.05	0.1253	0.1338	0.1287	0.1361	0.1312	0.1310
0.5	12	0.1	0.1207	0.1166	0.1185	0.1085	0.1106	0.1150
0.5	24	0.1	0.1356	0.1309	0.1393	0.1387	0.1307	0.1350
1	24	0.1	0.1545	0.1552	0.1374	0.1495	0.1473	0.1488

**Table 3 sensors-25-06863-t003:** Comparison of measurement results (unit: mm).

Parameter	Slicing Method	CMM	Absolute Deviation	Relative Deviation
Cylindricity	0.1210	0.1269	−0.0059	−4.6%
Straightness	0.0106	0.0111	−0.0005	−4.5%

## Data Availability

The original contributions presented in this study are included in the article. Further inquiries can be directed to the corresponding author(s).

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
