# Peer review of "Measurement of Form and Position Error of Small-Diameter Deep Holes Based on Collaboration Between a Lateral Confocal Displacement Sensor and Helical Scanning"

_sensors, 2025, doi:10.3390/s25226863_

Round 1

Reviewer 1 Report

Comments and Suggestions for Authors

The paper "Research on Form and Position Error Measurement of Small-Diameter Deep Holes Based on Collaboration of Lateral Confocal Displacement Sensor and Helical Scanning" presents an innovative method for measuring small-diameter, deep holes. The authors have developed a complex mathematical apparatus to justify the measurement method and deviations from cylindricity and straightness. Also, an experimental stand for measuring holes was created. Comparing the results obtained by determining the measurement errors led to the conclusion that the proposed method and apparatus are correct.
I appreciate that the paper is valuable and that it could be published in "Sensors", after minimal improvements by the authors.
Thus, the authors could clarify the scope of application of the method (minimum diameter, the ratio between the depth and diameter of the hole).
The authors could appreciate whether the method could also be applied to clogged holes.

Thank you,

Reviewer 2 Report

Comments and Suggestions for Authors

The article addresses a relevant problem of measuring the form and positional errors of small-diameter deep holes using a spectral confocal sensor and a helical scanning trajectory. The authors propose a new reconstruction algorithm (virtual slicing - B-spline reconstruction) and confirm its effectiveness through experiments. The work fits well within the scope of Sensors and has practical significance for precision metrology and mechanical engineering. However, the text contains a number of shortcomings that require revision.

  1. The article frequently contains tautologies and repetitions, for example: "lateral confocal spectral confocal principle" or "adaptive adaptive motion control algorithm." This reduces readability and requires editorial corrections.

  2. Sentences are often too long and overloaded with details, which makes comprehension difficult. It is recommended to simplify the syntax.

  3. In formula (1) for the measurement radius R=r+l+Δs it is not explained how the parameter l (measurement number) is determined, nor what the reference distance  means in practical terms.

  4. In Section 4.2, the results on cylindricity show a significant scatter (for example, from 0.0545 mm to 0.1552 mm under the same conditions). The authors attribute this to motion parameters, but more rigorous statistical justification is required.

  5. The comparison with CMM is limited to only one sample (∅6×200 mm). To strengthen the validity, more examples should be presented, possibly with different materials or diameters.

  6. The description of the "virtual slicing – B-spline reconstruction" contains many details, but lacks a comparative analysis with alternative methods (e.g., polynomial approximation, NURBS). It is not clear where exactly the advantages in speed and accuracy lie.

  7. Figure and table numbering is correct, but captions are too brief and do not explain the essence (for example, Fig. 6 "4th order B-spline fitting curve" does not clarify what is shown or how it relates to the results).

  8. The abstract contains an error: "adaptive adaptive motion control algorithm."

  9. Terminology is sometimes mixed: "straightness error" and "linearity error" are used as synonyms, which causes confusion.

  10. The tables present values with excessive precision (e.g., 0.0106 mm), although the actual accuracy of the experiment does not allow such digits to be guaranteed.

Comments on the Quality of English Language

The article frequently contains tautologies and repetitions, for example: "lateral confocal spectral confocal principle" or "adaptive adaptive motion control algorithm." This reduces readability and requires editorial corrections.

Reviewer 3 Report

Comments and Suggestions for Authors

The authors present a novel approach for measuring form and position errors in small-diameter deep holes by integrating lateral spectral confocal sensing with helical scanning and B-spline reconstruction. The proposed method addresses spatial constraints and data processing challenges inherent to traditional contact and non-contact techniques. 

  1. The manuscript appropriately cites references for the aerospace and hydraulic applications mentioned in the introduction. If the authors choose to expand the discussion to include additional application domains (such as medical devices, automotive components, or microfluidic systems), it is recommended that each new example be supported by a relevant citation.
  2. The manuscript describes the use of a Ø6 mm × 200 mm test piece for experimental validation, which appears to have been fabricated specifically for this study. To clarify the practical relevance of the results, please state explicitly whether this test hole corresponds to a real industrial component or was manufactured solely for validation purposes.
  3. Given the technical nature of the measurement setup, readers may benefit from a visual reference showing the physical appearance of the test hole under inspection. Consider including an image (either from a microscope or a macro lens) that illustrates the entrance and geometry of the deep hole. This would enhance accessibility and help contextualize the measurement challenge.
  4. Throughout the full text, figure references alternate between “Figure” and “Fig.” To ensure consistency and alignment with the editorial style the journal.
  5. The manuscript describes the reconstruction and evaluation process in detail but does not specify the computational time required to execute the algorithm. Additionally, it is unclear whether the results are visualized through a dedicated graphical interface or require external software for analysis. To enhance reproducibility and practical understanding, please include information on processing time and the visualization environment used to generate Figures 6–9.
  6. It is presented a detailed reconstruction method based on fourth-order B-spline fitting, which appears well-suited for processing dense helical point clouds. However, the rationale for selecting B-splines over other curve-fitting techniques is not explicitly discussed. Including a brief justification or comparison would clarify the design choices and strengthen methodological transparency.
  7. The manuscript states that the algorithm filters out noise and outliers caused by environmental factors such as illumination variability and mechanical vibration. However, it does not specify the filtering techniques used, nor does it quantify their effectiveness. Please include a concise explanation of the filtering method applied and, if possible, provide a brief quantitative or graphical assessment of its performance.
  8. The validation results focus on a single nominally straight and regular deep hole. To further demonstrate the robustness and sensitivity of the proposed method, consider including (or briefly discussing) measurements of test holes with intentionally introduced form deviations. Comparing such cases (for example, in the straightness evaluation shown in Figure 8) would provide stronger experimental evidence of the method’s capability to detect and quantify known geometric errors.

Comments on the Quality of English Language

The English is generally clear and understandable. However, minor language polishing is recommended to improve fluency and consistency. In particular, the manuscript would benefit from uniform terminology (e.g., “Figure” instead of “Fig.”)

Reviewer 4 Report

Comments and Suggestions for Authors

This manuscript presents a new non-contact method for measuring shape and position errors in holes with large aspect ratios (≥ 20) and small diameters (≤ 10 mm). The sensor head employs spectral confocal microscopy for displacement sensing. Dense helical 3D point cloud measurements of the internal surfaces of a Ø6 mm diameter and 200 mm deep hole are generated through motorized rotational (12-24 rpm) and axial (0.5-1 mm/s) motion. Subsequently, these measurements are processed with a virtual slicing-B-spline algorithm to estimate the hole's straightness and cylindricity errors. The presented results indicate that this method outperforms a commercial coordinate measuring machine (CMM) by approximately 4.5%.

I believe the subject of this manuscript fits well within the scope of Sensors due to its relevance to emerging precision measurement technologies for modern manufacturing and addresses challenges like spatial constraints and high-density data processing. However, the technical rigor and the presentation requires substantial refinement. Therefore, I recommend publication with major edits. While the technical and presentation aspects are intertwined in some areas, I have bifurcated my comments and questions loosely into two sections: (1) technical and (2) presentation.

1. Technical:

This section covers technical questions and comments requesting elaboration of details crucial for potential reproduction of the authors' results and/or follow-up studies by other researchers. I also seek clarification on suspected errors and their implications. The authors should verify and correct any identified errors or informational shortcomings to improve the manuscript's rigor and completeness.

(a) Further details in Sec. 2:

Sec. 2 currently dedicates a significant portion to the spectral confocal displacement sensor's operating principles but lacks sufficient detail on the axial-rotational synergistic motion mechanism. Further details are needed for two reasons:

- The tight motional tolerances for the helical measurement make it challenging to maintain adequate parallelism and clearance between the carbon fiber tube and the pipe fitting. The photograph in Fig. 4 shows an ordinary 3-jaw chuck to secure and rotate the pipe fitting. Please provide tolerances on the truing of the chuck. Any residual runout should influence straightness and cylindricity error measurements.

- A meaningful comparison between the authors' instrument and the professionally engineered Qingdao Leader CMM (discussed in Sec. 4.2.2) is only possible if the former demonstrates comparable or superior motion performance to the latter. While the linear stage appears to be a commercial product (please provide manufacturer and part number), my primary concern lies with the modest-looking rotation-producing setup. Beyond runout, the rotational motion must also account for critical factors such as hysteresis, backlash, and encoder errors. Please provide detailed specifications about the rotational motion.

The authors highlight the sub-micron accuracy and non-contact nature of the spectral confocal displacement sensor as a significant improvement over CMMs. However, if motion control acts as the bottleneck for accuracy and precision, then the optical sensor offers no real advantage. While the instrument's reported outperformance of the CMM suggests that motion control may not be the primary limitation, hard evidence is needed to convince a skeptical reader.

To address these concerns, I recommend the following:

- Divide Sec. 2 into at least two distinct subsections: one dedicated to the sensor and the other to the motion control.

- For the sensor, please provide detailed specifications. If it is a commercial device (e.g., Micro-Epsilon), specify the manufacturer and part number. If custom-manufactured, provide details on its construction, as miniaturizing the sensor (≲ Ø4 mm) with commercial optics (e.g., Thorlabs) seems non-trivial.

- Conduct a thorough analysis of all measurement and actuation errors, presenting this information clearly in a table. Identifying bottlenecks will inform future improvements.

- Clearly distinguish between the precision and accuracy of your instrument.

(b) Fig. 1 and Eqs. (1) & (19):

The inset in Fig. 1 is intended to define the variables (e.g., D, δ, Δs) denoting various distances in the apparatus and to visually illustrate their geometrical relationships. However, when the information in Fig. 1 and Eqs. (1) and (19) is taken together, there appears to be geometric inconsistencies and/or missing crucial information. These are some of the issues:

- Line 137 mentions "... measurement number 𝑙 fed back from the sensor … " and '𝑙' also appears in Eq. (1). A zoomed-in view of Fig. 1 suggests '𝑙' might denote the distance between the focus of blue rays and the virtual focus (dashed gray lines) of the red rays. If my interpretation is correct, then the condition in Eq. (1) does not seem to be satisfied.

- In Eq. (19), the factor of 2 in the "D/2" term on the right-hand side appears to be a typo. Ignoring this factor, 'δ' seems to represent the total gap between the carbon fiber tube and the inner walls of the test piece's hole. However, Fig. 1 depicts 'δ' as the gap on only one side.

These ambiguities and inconsistencies propagate to other sections of the manuscript, such as the discussion of the minimum measurable hole size in lines 311-314.

To address these concerns, I recommend the following:

- Recreate Fig. 1 using vector graphics to ensure text sizes and line art are clearly discernible on standard paper sizes (A4 or letter).

- Harmonize Fig. 1 with Eqs. (1) and (19) to resolve the identified discrepancies.

(c) Data preprocessing:

- In Sec. 3.1, the authors properly motivate the need for denoising by attributing noise sources to factors like local illumination, temperature drift, mechanical micro-vibrations, etc. However, the justification for the chosen filters (e.g., db4 wavelet) is not sufficiently explained. Moreover, the intuition behind the soft thresholding scheme in Eq. (2) is not clear.

- In Fig. 5, it is unclear which measurement value is plotted. Although the algorithm in Section 3.2 can accommodate hole axes oriented in any direction, the authors' specific setup aligns it with the z-direction. The optical sensor only measures radial distance (r), from which x and y are computed as x = r cos(θ) and y = r sin(θ). The z- and θ-coordinates simply reflect the linear and rotational actuators' encoder values respectively. If my understanding is correct, do the authors denoise only (x, y)? The plot in Fig. 5 would look very different if the z coordinate was also denoised.

- Is the signal "y(t)" explicitly represented as a function of time "t"? The manuscript does not explicitly define the parameter "t".

To address these concerns, I recommend the following:

- Please provide a brief justification for selecting specific denoising tools without an extensive signal processing explanation.

- Consider elaborating on the soft thresholding scheme in Eq. (3) or moving it to the appendices. In the current form of the manuscript, Eq. (3) distracts from the main narrative.

- Comment on which coordinates are denoised and justify your choice.

- Given the algebraic density of Sec. 3, ensure notational consistency between Secs. 3.1 and 3.2. Instead of "y(t)," consider "y(t_i)," where t_i = i Δt and the Δt = 0.1 s. This would be less confusing to the reader when relating "y(t_i)" to the point cloud {P_i}.

(d) Virtual slicing-B-spline reconstruction method:

Most of my comments on this subsection are in the presentation section below. Despite most likely being a typo, I believe its misleading nature qualifies it as a major technical error.

In step 8, I believe the signs of the inequalities are reversed in Eqs. (16) and (17). Please confirm this potential error.

(e) Cantilever deflection:

When discussing Coriolis and centrifugal forces in Eq. (20), a citation to Ref. [22] is provided. I believe this is a citation error. That reference only discusses the de-noising algorithm.

As mentioned earlier, some of the problems (specifically in lines 311-314) with Sec. 4 are related to Eq. (19). Putting aside those issues, I will address some other problems. For example, in Eq. (20), the mass of the probe (m) is not specified. Using the value of y_total = 0.08044 mm provided in the manuscript, I estimate m ~ 20 g.

Furthermore, Eq. (20) fails to account for the distributed mass of the T700 carbon fiber tube (m_t). Assuming a density of 1.8 g/cm³ and using the given dimensions, I estimate m_t to be around 4.5 g. After a quick lookup of basic beam deflection formulas, I believe this can be rectified by modifying Eq. (20) to replace m → m + (3/8) m_t.

According to my calculations, the contributions to y_total from the probe & tube mass and the centrifugal & Coriolis forces should be 87.66%, 7.50%, 4.93%, and 0.01%, respectively. Due to the issues previously mentioned in lines 311-314, I am unable to determine how this correction impacts the overall conclusions. Nevertheless, I strongly recommend that the authors implement this fix to ensure rigor and completeness.

(f) Data and interpretation for straightness and cylindricity errors:

Currently, the discussion of straightness and cylindricity measurements (lines 365-394) lacks sufficient coherence and hinders a complete understanding of the data and interpretations. This is primarily due to missing information on the sampling rate and an insufficient systematic isolation of the software (algorithm) from the hardware (vibrations) components of the sensor.

Did the authors change the sampling rate from 10 Hz in addition to the rotation and feed speeds? It would be beneficial to exploit the higher sampling rates possible with non-contact optical sensors. Assuming the sampling rate was maintained at 10 Hz, and a fixed hole depth, the total number of data points (N) should depend solely on the feed rate, not the rotation speed. Of course, both rotation and feed speeds would influence the axial (Δz) and angular (Δθ) sampling intervals within the point cloud.

The current discussion in lines 365-394 lacks systematic arguments on how changes in N, Δz, and Δθ affect the virtual slicing-B-spline reconstruction algorithm. Therefore, the authors' statements such as "… the assessment of straightness has a low demand for point cloud density …" (lines 378-379) are unsubstantiated. The impact of N, Δz, and Δθ on the algorithm can be easily investigated using a simulated point cloud. This would eliminate ambiguity from practical imperfections like cantilever beam vibrations.

When discussing straightness error (lines 365-380), the authors state that the centrifugal force is negligible while the Coriolis force is not. This contradicts the percentage contributions to y_total (Eq. (20)) I listed above. Please explain this apparent discrepancy. However, I think the authors' assertion that vibrations at higher rotation speeds negatively impact straightness error measurements sounds plausible.

The aforementioned simulations could be empirically corroborated by lowering rotation speeds to a point where vibrations are not a concern. This shouldn't be too difficult given the ω² scaling of the centrifugal force term. Such tests would better illustrate how these three parameters need to be optimized differently for straightness and cylindricity measurements.

Addressing the above comments would significantly enhance the rigor of lines 365-394.

(g) Comparison to commercial CMM:

Comparison with a commercial CMM is a crucial benchmarking step. Mention of the manufacturer (Qingdao Leader) and model (NCA686) of the CMM offers a useful reference for readers.

For completeness, Sec. 4.2.2 should elaborate on the CMM measurements. This includes specifying the manufacturers and models of the probe, stylus, and stylus extension. Please specify the orientation of the pipe fitting (horizontal or vertical) during CMM measurements and whether a vertically oriented spectral confocal sensor could mitigate cantilever deflections in future work.

Additionally, please include the following details:

- How were the CMM measurements segmented?
- Do the CMM scans also employ a segmented helical trajectory or an alternative method? 
- How much of the data processing was done by the CMM's software and how much post-processing was required?

Furthermore, comments on how the two measurement methods compare based on the surface roughness of the hole's inner surface would be beneficial to readers. In principle, both contact with and optical reflectance from rough surfaces should have some impact on the measurement.

Including these additional details in no more than a 300-word paragraph would suffice.

2. Presentation:

The manuscript is logically structured. However, its readability is significantly hampered by numerous technical typos, confusing notation, and unclear figures. I believe the following edits would greatly enhance the readability of the manuscript.

(a) Figures:

- In general, please make all figure captions a bit more descriptive and use vector graphics wherever applicable.

- Please improve the resolution of Fig. 3. Make the lines inside the cylinder to denote trajectories and annotations thicker.

- Please improve the resolution of Fig. 5 and possibly zoom in a bit. The pre- and post-noise curves are indiscernible.

- In Fig. 6, it might be more useful to include a 2D plot of either the x or y coordinates, at a few selected angular locations, showing both the fitted spline and raw data points as a function of z.

- In Fig. 8, it looks like the minimum inclusive cylinder is not parallel to the z axis. If so, please specify its axial vector. As I discuss under subsection (b) below, it would help the reader to better visualize step 7 of your virtual slicing-B-spline reconstruction algorithm if you include a sketch showing L_axis, and vectors v and A.

- In Fig. 9, except "measurement point," it's hard to locate the entities the legend is supposed to describe. The acronym "MIC" is undefined, leading to ambiguity regarding whether it represents the minimum or maximum inscribed cylinder. For a first-time reader, it's difficult to visualize the concepts of minimum and maximum inscribed cylinder based solely on the description in lines 255-259 and 266-270. Adding two more panels to Fig. 9 schematically sketching what's described in those lines would greatly aid the readers.

- I believe the "Parameter column" in the first row of Table 3 should say "Cylindricity."

(b) Virtual slicing-B-spline reconstruction algorithm:

- In step 1, the least squares cylindrical fitting method has not been detailed or cited.

- In step 3, in spite of the citation to Ref. [23], please briefly define the parameters "k", "n" and "u" to make the manuscript self-contained.

- In step 6, label the sub-enumerated list items as (a)-(f). Currently, it seems like steps 7 and 8 are part of the sub-enumerated list.

- In step 7, please simplify the explanation of minimum inclusive cylindrical optimization and explain the newly-introduced variables more clearly. As stated earlier, a sketch as a panel of Fig. 8 would greatly aid a first-time reader since the vector formulas are hard to visualize. Explicitly state that L_axis belongs to the minimum inclusive cylinder. Based on your description, it's not clear A represents the end point of a line connecting Q_k to L_axis and that your nonlinear constrained gradient optimization is trying to minimize the length of this line. Since you label the fitting circles as (u_c, v_c), please avoid reusing "v" to label the direction vector of the line L_axis. Due to the dense mathematical nature of Ref. [25], the readers' reliance on this citation should be reduced. Please remove Eq. (14) as it distracts from the flow of the text.

(c) Table of setup parameters:

Important parameters relevant to the authors' setup are scattered throughout the manuscript. Some examples of these include inner (d) & outer (D) diameter of the carbon fiber tube, rotation frequency (ω), feed speed (v), sampling interval (T). It would greatly help the reader to have all these quantities, the variables used to denote them, and their nominal values in a centralized location like a table. This table would especially be useful when looking at Eq. (20).

Round 2

Reviewer 2 Report

Comments and Suggestions for Authors

The authors have taken all my comments into account, the article can be accepted for publication.

Reviewer 4 Report

Comments and Suggestions for Authors

Thank you for addressing my comments and questions. The revised draft has corrected some of the major technical errors (e.g. sign of inequalities) and improved the presentability (e.g. figure updates) of the manuscript material. Moreover, the technical information added by the authors (e.g. CMM details) should aid other researchers interested in building on and/or reproducing the authors' work.

I have a minor suggestion. After rectifying Eq. (1), and eliminating the variable "l", I don't think this variable serves any purpose in Fig. 1. I recommend removing that annotation to avoid confusing the reader.

In summary, I recommend acceptance with minor edits.
